# Statins Effects on Blood Clotting: A Review

**DOI:** 10.3390/cells12232719

**Published:** 2023-11-27

**Authors:** Carmine Siniscalchi, Manuela Basaglia, Michele Riva, Michele Meschi, Tiziana Meschi, Giampiero Castaldo, Pierpaolo Di Micco

**Affiliations:** 1Angiology Unit, Department of Internal Medicine, Parma University Hospital, 43121 Parma, Italy; 2Department of Internal Medicine, Parma University Hospital, 43121 Parma, Italy; 3UOC Internal Medicine, Fidenza Hospital, 43036 Parma, Italy; 4Department of Medicine and Surgery, Parma University Hospital, 43121 Parma, Italy; 5AFO Medicina PO Santa Maria delle Grazie, Pozzuoli Naples Hospital 2 Nord, 80078 Naples, Italy

**Keywords:** venous thromboembolism, statin, tissue factor, thrombin, fibrinogen, fibrin, factor V/Va, factor XIII/XIIIa, protein C pathway, tissue factor pathway inhibitor

## Abstract

Statins are powerful lipid-lowering drugs that inhibit cholesterol biosynthesis via downregulation of hydroxymethylglutaryl coenzyme-A reductase, which are largely used in patients with or at risk of cardiovascular disease. Available data on thromboembolic disease include primary and secondary prevention as well as bleeding and mortality rates in statin users during anticoagulation for VTE. Experimental studies indicate that statins alter blood clotting at various levels. Statins produce anticoagulant effects via downregulation of tissue factor expression and enhanced endothelial thrombomodulin expression resulting in reduced thrombin generation. Statins impair fibrinogen cleavage and reduce thrombin generation. A reduction of factor V and factor XIII activation has been observed in patients treated with statins. It is postulated that the mechanisms involved are downregulation of factor V and activated factor V, modulation of the protein C pathway and alteration of the tissue factor pathway inhibitor. Clinical and experimental studies have shown that statins exert antiplatelet effects through early and delayed inhibition of platelet activation, adhesion and aggregation. It has been postulated that statin-induced anticoagulant effects can explain, at least partially, a reduction in primary and secondary VTE and death. Evidence supporting the use of statins for prevention of arterial thrombosis-related cardiovascular events is robust, but their role in VTE remains to be further elucidated. In this review, we present biological evidence and experimental data supporting the ability of statins to directly interfere with the clotting system.

## 1. Introduction

Venous thromboembolism (VTE) is a worldwide cause of morbidity and mortality [1] and represents a major health problem due to its prevalence, incidence and costs [2]. Low molecular weight heparin (LMWH), unfractionated heparin (UFH), vitamin K antagonists (VKA) and direct oral anticoagulants (DOACs) are the gold standard of treatment for VTE [3,4], despite their related risk of bleeding [3,4]. When anticoagulation is stopped in patients with unprovoked VTE, the risk of VTE recurrence is up to 10% per year [5] whereas the recurrence rate of VTE during anticoagulation is 3% in the first 3 to 6 months [4,6]. A drug that reduces the risk of VTE recurrence without increasing the risk of major bleeding and mortality is hypothetically the ideal therapy. Several studies have been conducted on statin use during anticoagulation for VTE. Statins are powerful lipid-lowering drugs that inhibit cholesterol biosynthesis via downregulation of hydroxymethylglutaryl coenzyme-A reductase [7]. Statins are largely used in patients with or at risk of cardiovascular disease, due to their protective role in myocardial infarction, stroke, and cardiovascular death [8]. A number of ’pleiotropic’ effects of statins, independent of blood cholesterol reduction, have been proposed to explain their efficacy in cardiovascular disease [9,10].

Data are available on thromboembolic disease regarding primary [11,12] and secondary prevention [13,14,15,16,17,18] as well as bleeding [19,20,21,22,23,24,25] and mortality rates in statin users during anticoagulation for VTE [26,27,28,29].

In vitro studies and animal models have demonstrated that statins [30,31,32,33] affect platelet aggregation. Statins affect P selectin levels and platelet aggregation [34] and also modulate platelet-derived NO release in a dose-dependent manner and independently of cholesterol-lowering effect [35]. Statins are able to affect some components of blood clotting such as tissue factor (TF) gene expression [36,37], von Willebrand factor [38,39], D-dimer levels [39], and coagulation factors VIII, VII and XI [40].

In this review, we present the biological evidence and experimental data supporting the ability of statins to interfere with the clotting system and platelets and their postulate anticoagulant activities.

## 2. Statin Effect on Tissue Factor

Tissue factor (TF), an integral membrane glycoprotein [41], is constitutively expressed on surfaces of skin, organs, vascular adventitia, and many of their malignant counterparts, but it is not normally expressed on vascular endothelial cells or monocytes. TF is exposed to blood flow only after endothelial damage [42]. The majority of TF is in a functionally inactive (“encrypted”) state. Upon cell lysis, TF will become activated (“de-encrypted”) and support FVIIa binding and the activation of factor X. Colli et al. [36] demonstrated that fluvastatin and simvastatin dose-dependently inhibit the activity of tissue factor (TF) [43].

A decrease in TF mRNA expression and activity through the inhibition of transcription nuclear factor κB (NF-κB) was observed during simvastatin and fluvastatin treatment [36]. Similarly, in cerivastatin, atorvastatin, simvastatin, pravastatin, lovastatin, and fluvastatin users, a reduction in TF mRNA expression in endothelial and vascular smooth muscle cells was observed [44]. A reduction in TF activity and expression was also observed in hypercholesterolaemic patients during statin therapy [45] and in aortic smooth muscle and endothelial cells during statin therapy [37]. Activation of “Protease Activated Receptors” (PAR) by thrombin induces TF expression in endothelial cells. Banfi et al. [46] demonstrated that rosuvastatin and fluvastatin block the induction of TF expression by PAR1 relocation and inhibition of ERK1/2 phosphorylation, probably due to a decrease in cell membrane cholesterol [46]. In atherosclerotic plaques removed during endarterectomy, 29% lower TF antigen levels were observed, and 56% lower TF activity was observed in atorvastatin users compared with a placebo [47]. A similar statin-induced decrease in TF expression has also been observed in atherosclerotic plaques removed from coronary arteries [48].

The capacity of statins to induce downregulation of TF, independently to their cholesterol-lowering effect, has also been observed in animal models [49,50,51,52]. In experimental studies on LDL receptor-deficient mice treated with a four-week high-fat diet, the use of simvastatin decreased thrombin generation, oxidized LDL (oxLDL), white blood cell TF expression and the number of TF+ microparticles. Evidence for statin-induced downregulation of TF expression also comes from prostate cancer studies. Prostate cancer cells are capable of releasing secretory granules rich in cholesterol that are TF bearing with pro-coagulant activities. Simvastatin seems to be capable of reducing the pro-coagulant activities of these secretory granules [53].

These data were confirmed in patients affected by polygenic hypercholesterolemia [45] and in subjects with low density lipoprotein (LDL) cholesterol >130 mg/dL [54]. Furthermore, in vitro studies confirmed that the prevention of isoprenoid intermediate synthesis by statins is crucial for TF inhibition. Statins downregulate the expression of TF via inhibition of geranylgeranylation of the Rho/Rho kinase pathway, an enzyme that upregulates the expression of TF in cultured endothelial cells and monocytes via nuclear factor-κB activation [37,55,56,57].

## 3. Statin Effect on Thrombin

Thrombin is the main initiator of physiological blood clotting and pathological thrombosis [58]. Thrombin is the cofactor for activated FVII (FVIIa) [59,60] and is also implicated in FV activation, fibrinogen clotting, platelet activation, crosslinking transglutaminase FXIII activation, endothelial cell activation, enhanced TF expression, cell proliferation and vascular constriction. Thrombin is inactivated by antithrombin with production downregulated by the dynamic protein C (PC) system.

Several clinical and experimental data, derived from a variety of in vitro and in vivo studies, indicate that statin therapy results in decreased thrombin formation [61,62,63,64]. Pravastatin 15 mg/die has been effective in reducing platelet-dependent thrombin generation in hypercholesterolemic subjects [65]. Simvastatin from 20 to 40 mg/die depresses thrombin generation in subjects with hypercholesterolemia [66,67,68]. Simvastatin use is associated with a marked inhibition of thrombin generation in plasma obtained from healthy individuals and hypercholesterolaemic patients [69]. In these subjects, a 16% reduction in the rate of prothrombin depletion and 27% decrease in the rate of formation of prothrombin activation products [68] with a longer duration of the initiation phase of blood coagulation was observed [68]. These effects were independent of the pre- or post-treatment blood cholesterol concentrations [68].

Within the first 12 h of acute myocardial infarction, statin use is associated with lower thrombin generation and platelet activation [70]. On the contrary, in acute coronary syndromes (ACS) in which the rupture of a coronary atherosclerotic plaque enhanced blood thrombogenicity [71], studies using high-dose atorvastatin did not demonstrate a reduction in thrombin generation [72]. In patients with a high cardiovascular risk and high LDL cholesterol, simvastatin is capable of lowering the total thrombin generated at the site of microvascular injury by 30%, independently of cholesterol reduction [73]. Similar effects were observed after one month of atorvastatin administration, with rates of thrombin formation decreased by 34–40% [74]. Similarly, simvastatin reduced the rate of thrombin generation triggered by vascular injury in hypercholesterolaemic patients [68].

Reduced thrombin generation was also observed in elderly patients with atrial fibrillation undergoing oral anticoagulation [75], those with peripheral artery disease [76] and after percutaneous coronary intervention [77].

Statins are effective in reducing thrombin generation even in type 2 diabetes patients [78]. Studies demonstrated a significant reduction in thrombin production in diabetic patients using statins without prior cardiovascular events [79,80].

It is important to note that the effect of statins on different subjects seems to be genetically determined. In particular, the +5466 G allelic variant of the TF gene is directly responsible for statin action and occurs in 16% of Europeans. In carriers of the +5466 G allele, a greater decrease in thrombin formation is observed after administration of simvastatin for three months compared to subjects with the +5466 AA genotype [81].

In conclusion, several studies strongly suggest that statins are able to reduce thrombin formation. Thrombin activities are implicated in several atherogenic processes, such as cell migration/proliferation, leukocyte trafficking, and inflammation [82]. For this reason, it is plausible to speculate that statin-induced reduction in thrombin formation might also have various indirect mechanisms to reduce thromboembolic events [82].

## 4. Statin Effect on Fibrinogen and Fibrin

Fibrinogen is a principal substrate for thrombin. Thrombin catalyzes the release of fibrinopeptide A (FPA) and B (FPB) to form a fibrin monomer that can polymerize with other fibrin monomers, resulting in double-stranded fibrils that associate laterally, forming thick fibers [83,84]. Elevated fibrinogen levels represent a risk factor for thrombosis and the effects of statins on fibrinogen concentrations is a matter of debate. Statins impair fibrinogen cleavage, with reduced thrombin generation. In hypercholesterolaemic subjects after three-month treatment with simvastatin, decreased amounts of fibrinopeptides have been reported, independent of cholesterol reduction [66,67]. Statins are able to alter the characteristics of the fibrin clot. In a study by Undas et al., treatment with statins led to an increase in the permeability and a more rapid lysis of the fibrin clot [85]. Statins may also modulate the fibrin clot phenotype in young or middle-aged patients with previous VTE, as reported in a study in which atorvastatin increased fibrin clot permeability and lysability, regardless of its lipid-lowering capacity [86].

## 5. Statin Effect of Factor V/Va

Factor V is activated by thrombin. A decrease in thrombin generation or function leads to a decrease in FV activation. Undas et al. demonstrated that after three months of simvastatin, the formation of FVa at the site of microvascular damage was reduced, regardless of simvastatin-induced changes in lipid profiles [68]. In the same study, it was observed that FVa reduction was associated with a lower rate of FV depletion [68]. Two studies demonstrated that atorvastatin [74] and simvastatin [54] reduced FV in cardiovascular patients [74].

## 6. Statin Effect on Factor XIII/XIIIa

To our knowledge, only few data are available on the role of statin in Factor XIII. A reduction of 20% of FXIII activation was observed in patients treated with simvastatin [68]. Thrombin is the major FXIII activator in vivo and this effect most likely reflects decreased thrombin formation observed during statin administration.

## 7. Statin Effect on Protein C Pathway

There are several inhibitors that can suppress thrombin formation. These include heparin cofactor II, antithrombin, tissue factor pathway inhibitor, and protein C pathway [59,87]. The so-called “protein C pathway system” is composed of protein C, protein S and thrombomodulin (TM) [88]. TM is able to generate a complex with thrombin which loses its procoagulant activity, transforming into an “anticoagulant” enzyme [88]. Data on the expression of TM during statin therapy are scarce. Several studies have demonstrated lower levels of TM in the plasma of hypercholesterolemic subjects treated with pravastatin [89] and in subjects undergoing heart transplantation after treatment with fluvastatin [90]. Other studies, however, failed to demonstrate any significant statin-induced changes in TM levels [91].

## 8. Statin Effect on Tissue Factor Pathway Inhibitor

Tissue factor pathway inhibitor (TFPI) is the only physiological inhibitor of the TF-FVIIa complex [92]. TFPI is mainly bound to LDL. For this reason, it is assumed that drugs capable of altering the blood lipid profile can potentially alter the activity of this inhibitor. However, studies are currently inconclusive on this matter [61,63]. It has been reported that lovastatin [93], fluvastatin [94], simvastatin [95], and atorvastatin [95] are able to decrease the activity of TFPI through a reduction in LDL-TFPI complexes and total TFPI levels without any changes in free TFPI. However, the available data indicate a lack of significant changes in TFPI levels and/or activity during statin therapy.

## 9. Factor VII

The capacity of statins to modulate FVIIa activity and levels is a matter of debate. In hypercholesterolemic patients, factor VII antigen levels and FVII coagulant activity have been reported to be lower (12%) during treatment with atorvastatin [96,97] independently to cholesterol lowering effects mediated by this agent. In an in vitro study, FVIIa activity remained unaltered after statin therapy [96]. Higher doses of atorvastatin, administered for 12 weeks in hyperlipidemic subjects, reduced FVII coagulant activity by 8% [98].

However, studies have reported that simvastatin [99], pravastatin [100], and fluvastatin [101] seem to have no effect on FVII levels. Further studies conducted specifically on this topic and using specific assays for Factor VIIa are needed.

## 10. Effect of Statin on Platelets

Clinical and experimental studies have demonstrated that statins possess antiplatelet effects mainly on activation, adhesion and aggregation.

These effects can be summarized as follows:reduced activation of NOX2-derived oxidative stress induced by statins may increase NO activity and synthesis [102];statins are able to increase NO synthase activity via platelet cyclic guanosine monophosphate [103];downregulation of nicotinamide adenine dinucleotide phosphate oxidase-2 (NOX2), with reduced activation of NOX2-derived oxidative stress and reduced formation of proaggregatory platelet isoprostanes [104];inhibition of thromboxane A2 formation supports both an early antiplatelet effect of statins, through reduced activation of cyclooxygenase-1, and a late-dependent hypocholesterolemic effect [104,105];statins have the ability to inhibit platelet thrombin protease-activated receptor 1 with inhibition of its pro-thrombotic properties [106];statins can downregulate two potent platelet agonists: platelet membrane expression of CD36 and oxLDL lectin-like receptor 1 [105];statins can inhibit the collagen-induced platelet ligand CD40, impairing the interaction of platelets with the endothelium [107];statin therapy reduces plasma endothelin-1 concentrations [108].

A study on hypercholesterolemic subjects demonstrated that pravastatin affects platelet-dependent thrombin generation and platelet aggregation [31,32,33]. P selectin levels and platelet aggregation were also reduced by atorvastatin, simvastatin, fluvastatin, and cerivastatin [34]. They also modulate the platelet-derived NO release in a dose-dependent manner and independently of a cholesterol-lowering effect [35]. A list of the putative effects of statin treatment on platelets is reported in Table 1.

## 11. Clinical Benefit and Clinical Risk

The clinical role of statins during anticoagulation for VTE has been investigated in studies on primary and secondary VTE prevention.

The first evidence of statins’ antithrombotic properties comes from the JUPITER study, a large placebo-controlled randomized trial that reported a 43% reduction in the first episode of VTE in rosuvastatin users [11]. These findings were confirmed by four cohort studies and four case-control studies [109].

Data are also available on statins’ secondary prevention of VTE during anticoagulation. A lower risk of recurrent VTE was reported: (1) in a secondary analysis of the EINSTEIN VTE [15]; (2) in a retrospective case-control study [16]; (3) in a large study on elderly patients; (4) in a prospective cohort study in Denmark [13]; (5) in two Danish population-based registries [13]; (6) in a Dutch registry [14]; (7) in a sub-analysis of a study from RIETE registry [26].

There are conflicting data on the rates of major bleeding promoted by statins during anticoagulation for VTE. A non-significant reduction in major bleeding rates was found in a post hoc analysis of the EINSTEIN DVT and PE program [15]. Studies reported an increased risk of bleeding in statin users during anticoagulation with Dabigatran [20] and during stroke prevention by an aggressive reduction in cholesterol levels (SPARCL study), in which stroke risk reduction was associated with an increased risk of hemorrhagic stroke [21].

However, contrasting conclusions come from other studies [11,22,23,24]. In “The Heart Protection Study,” similar rates of hemorrhagic strokes were found in statin users and non-users [22]. In a large study of patients with VTE, statin users had a similar rate of major bleeding compared to non-users [26]. Similar results were found in the Intervention Trial Evaluating Rosuvastatin (JUPITER) [11] and in a meta-analysis of randomized trials and observational studies [24]. However, as stated by the American Heart Association/American Stroke Association guidelines [25], no recommendation is made against the use of statins during intracerebral hemorrhage.

Only a few studies have investigated the influence of statins on mortality in patients undergoing anticoagulant therapy for VTE, suggesting that statin users had a lower risk of death than non-users [14,18,26]. It seems plausible that a lower mortality rate is independent of the lipid-lowering effect of statins and rather related to their “pleiotropic” effects on inflammation, platelet inhibition, increase in nitric oxide production and downregulation of the coagulation cascade [30].

To date, evidence favors a protective role of statins for all-cause death and VTE recurrence [18,26,27,28,29,102]. However, intervention studies specifically designed for this topic are needed and these findings should be taken with caution.

Despite the postulate benefits of statins in VTE patients, they also have the potential to cause side effects. The two best documented side effects in observational studies and clinical trials are an increased risk of myopathy and an increased incidence of diabetes. Other side effects, such as the potential to impair memory and cognition, promote cataract formation, and/or compromise kidney outcomes have been proposed but not convincingly demonstrated.

## 12. Discussion

Several studies suggest that statins are able to affect coagulation through different mechanisms. The cholesterol-lowering effect of statins is well known and their beneficial effect on cardiovascular risk is well studied. However, there is growing interest in the antithrombotic properties of statins that are not directly associated with changes in lipid profiles. These effects are enhanced by the profibrinolytic and antiplatelet properties of statins. The anticoagulant effects of statins are greater in subjects with hypercholesterolemia but they are not associated with the extent of cholesterol reduction because the anticoagulant effects of statins are observed 1–3 days after their administration.

The main anticoagulant effect of statins occurs through the downregulation of TF expression and the upregulation of endothelial TM expression, resulting in a reduction in thrombin generation and consequently a reduction in various thrombin-catalyzed pro-coagulant reactions, such as fibrinogen cleavage and activation of FV and FXIII, coupled with increased expression of thrombomodulin. It has been hypothesized that statin-induced anticoagulant effects may at least partially explain a reduction in primary and secondary VTE recurrences and death. Data are available on primary [11,12] and secondary [13,14,15,16,17,18] VTE prevention but also on bleeding [19,20,21,22,23,24,25] and mortality rates in statin users during anticoagulant therapy for VTE [26,27,28,29].

Furthermore, it remains uncertain whether all available statins share similar mechanisms of anticoagulant properties; for example, studies have demonstrated that rosuvastatin [30] and atorvastatin [31,32,33] influence platelet aggregation but reagrding other types of statins, data are conflicting.

A study on hypercholesterolemic subjects demonstrated that pravastatin affects platelet-dependent thrombin generation and platelet aggregation [31,32,33]. P selectin levels and platelet aggregation were also reduced by atorvastatin, simvastatin, fluvastatin, and cerivastatin [34]. They also modulate the release of platelet-derived NO independently of their cholesterol-lowering effect [35]. There is anecdotal evidence for the influence of statins on coagulation from as early as the 1990s and 2000s. One of the first pieces of evidence suggests that both simvastatin and fluvastatin were able to reduce tissue factor (TF) gene expression [36,37]. Lower plasma levels of von Willebrand factor [38,39] and D-dimer [39] have also been found in statin users in two meta-analyses of randomized trials [38,39] and in patients with previous VTE [40]. A similar decrease was observed in the levels of coagulation factors VIII, VII and XI during rosuvastatin therapy [40].

However, the anticoagulant effects of statins are difficult to translate into clinically relevant outcomes. First, the suppression of coagulation by statins could be explained by the cholesterol- or oxLDL-lowering effect. Secondly, patients invariably have reduced blood cholesterol and it is difficult to assess the relative strength of the anticoagulant effect versus the lipid-lowering action.

Finally, it is plausible to consider the antithrombotic effect of statins as “specific” due to their unique ability to interfere with both the coagulation cascade and platelet functions. A list of the putative effects of statin treatment on coagulation factors is reported in Table 2.

## 13. Conclusions

In conclusion, several studies have demonstrated that statins exert a dual anticoagulant–antiplatelet action and therefore have a role in the prevention of thromboembolic events. In detail, the anticoagulant effect of statins is determined by their ability to reduce the production of tissue factor (TF), by their ability to decrease the generation of thrombin and by a modulation of the majority of pro-coagulant reactions catalyzed by thrombin (i.e., the cleavage of fibrinogen, activation of factor V and factor XIII). The anticoagulant effects of statins are also mediated by an increased expression of endothelial thrombomodulin and increased fibrinolytic activity. In addition to these effects on coagulation, statins are able to exert a direct antiplatelet effect through immediate and delayed inhibitory mechanisms on platelet function. In addition to these direct antithrombotic actions, additional plausible properties of statins (i.e., anti-inflammatory and/or antioxidant) may indirectly interfere with thrombus formation and propagation.

Although statins influence the levels of a multitude of hemostatic factors in an antithrombotic direction, data supporting their use for venous thromboembolism prevention are not consistent, and the impact of statins on VTE is still debated. The protective role of statins in the prevention of cardiovascular disease is solid and well-studied. However, their postulated protective role against VTE is a matter of debate and needs to be further clarified by studies performed on this topic. However, their antithrombotic properties do not appear to be related to their lipid-lowering effects, but rather to their pleiotropic effects, in particular their anti-inflammatory, direct and indirect effects on coagulation and platelet aggregations. Although statin treatment is able to influence the levels of several coagulation factors, the possible beneficial clinical effects induced by statins in patients with VTE are currently under debate and need future studies to be reinforced. In particular, it has been postulated that statin-induced anticoagulant effects can explain, at least partially, a reduction in primary and secondary VTE and death. However randomized clinical intervention studies specifically designed on this topic are needed.

## Figures and Tables

**Table 1 cells-12-02719-t001:** Effects of statin treatment on platelets.

Substrate	Effect of Statins	References
Nitric Oxide	Up-regulation	[102,103]
NOX2 and isoprostanes	Down-regulation	[104]
COX-1 and thromboxane A2	Down-regulation	[104,105]
PAR-1 signaling	Down-regulation	[106]
CD36 and lectin-like oxLDLR-1	Down-regulation	[105]
CD40 ligand	Down-regulation	[107]
endothelin-1	Down-regulation	[108]
Platelet-Activating Factor (PAF)	Inhibitory effect	[109]

COX-1, cyclooxygenase-1; NOX2, nicotinamide adenine dinucleotide phosphate-oxidase 2; PAR-1, protease-activated receptor-1.

**Table 2 cells-12-02719-t002:** Effects of statin treatment on coagulation.

Substrate	Effect of Statins	References
Tissue factor (TF)	Decrease of TF mRNA expression and activity	[37,43,44,45,46,47,48,49,50,51,52,53,54,55,56,57]
Thrombin	decreased thrombin formation	[2,3,4,5,6,7,8,9,10,11,12,13,14,15,16,17,18,19,20,21,22,23,24,25,26,27,28,29,30,31,32,33,34,35,36,37,38,39,40,41,42,43,44,45,46,47,48,49,50,51,52,53,54,55,56,57,58,59,60,61,62,63,64,65,66,67,68,69,70,71,72,73,74,75,76,77,78,79]
Fibrinogen	statins impaired fibrinogen cleavage and increase plasma fibrin clot permeability and lysability	[66,67,85,86]
Factor V	Down-regulation	[54,68,74]
Factor XIII	Down-regulation	[68]
Tissue factor pathway inhibitor (TFPI)	Down-regulation of TFPI activity	[93,94,95]
Factor VII	Down-regulation/no effect	[96,97,98,99,100,101]

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
