# Peer review of "Statins Effects on Blood Clotting: A Review"

_cells, 2023, doi:10.3390/cells12232719_

Round 1

Reviewer 1 Report

Comments and Suggestions for Authors

The authors of the manuscript focused on the effect of statins on coagulation. Statins are a potential treatment for venous thromboembolism (VTE) prophylaxis complementary to conventional anticoagulants without associated bleeding complications. TThere is growing evidence that treatment with statins can lead to a significant downregulation of the blood coagulation cascade, most probably as a result of decreased tissue factor expression, which leads to reduced thrombin generation.

Page 4, lines 155-161: The authors should describe in more detail about fibrinogen. Fibrinogen has a characteristic trinodal organization, consisting of a single central and two distal nodes. Normal fibrinogen blood levels vary and are typically given as 1.8–4.2 g/L This statement was published in a manuscript which should be quoted: ,, J. Clin. Med. 2022, 11(4), 1083; https://doi.org/10.3390/jcm11041083.“

Authors should add figures or a table.

I have to say that with these 117 references, most of them are older than 5 years. It is also appropriate to add newer references

Author Response

Many thanks for your review
I modified the main text according to your suggestions.
In particular:
1. I have described fibrinogen in more detail and have included the sentences you suggested and the related quote
2. I produced 2 tables and 2 figures
3. I added more recent references

Best regards

Carmine

Reviewer 2 Report

Comments and Suggestions for Authors

In this review, the authors presented the biological evidences and the experimental data supporting of the ability of statins to directly interfere with the clotting system.
Please explain all abbreviations used, including in the abstract (eg VTE).
Your study seems to be a narrative review, but please specify clearly in the article what kind of review it is.
In your article you mention two tables, but they are not present.
The article presents 117 references, but these are not current. You have only one article from the year 2022. I recommend adding references from the last two years.
The article contains technical editing errors (e.g. the keywords are in a different type of writing).

Author Response

Many thanks for Your review

I corrected the manuscript according to your suggestions, in particular:

1.I have explained all the abbreviations used, even in the abstract. I have highlighted in yellow in the main text.
2. I added (highlighted in yellow) that this is a narrative review.
3. I added the two tables
4. I added references from the last two years
5. I corrected the technical editing errors (highlighted in yellow)

Best Regards

Carmine

Reviewer 3 Report

Comments and Suggestions for Authors

This is a well-written narrative review that provides a great amount of data and mechanisms.

The article can be improved with the addition of 1 or 2 figures that will present the effects of statins on clothing.

Is there evidence that the presence of CKD modifies the effects of statins on blood clothing? Please clarify this issue.

Author Response

Many Thanks for Your review.

  1. I added a figure explaining the effects of statins on blood clotting.
  2. I added the following sentences in the text to answer your question (thus citing two other articles from "Cells" and "Sci. Rep"): "Using the data acquired from the meta-analysis by Esmeijer et al. (121), it was determined that statins have a beneficial influence on patients with Chronic Kidney Disease (CKD). In this subset of patients, low doses of statins cause improvements in cellular oxidative status, while high doses exacerbate oxidative stress, ultimately leading to cell death. Statins in normolipidemic patients with CKD reduce mortality regardless to their lipid lowering effect. (122).  For this reason, it is plausible to postulate that statins are able to exert their role on blood coagulation also in patients with CDK according to the mechanisms already mentioned.

Reviewer 4 Report

Comments and Suggestions for Authors

Statins effects on blood clotting: a review

General comments and issues:

in a narrative review, the authors present biological evidence and experimental data supporting biological/pharmacological interference of statins with the clotting system and platelet action and thereby independently and specifically supporting "anticoagulant activities".  This topic is of high clinical interest. However, there are some major issues with respect to the actual selection and presentation of the data.

Special comments and issues:

--> the selection criteria of the the scientific publications being cited in this review are unclear. Therefore, a significant selection bias cannot be ruled out. Please provide keywords and systematic procedures (e.g. flow chart) on how the cited publicatons have been selected. Importantly, potentially contradictory results and data also must systematically be discussed

--> the literature being cited and discussed in the text must be clearly summarized in comprehensive tables including the major items of the publication of interest like "clinical study", "experimental study", population or biological material under investigation, interventions being made, controls and major outcomes. The reader must be enabelled to quickly read and understand the major sources of evidence, the results of interest and major scientific limitations.

--> the manuscript must include some comprehensive illustrations and flow charts presenting the potential pharmacological mechanisms of interaction e.g. with Thrombin, Fibrinogen/Fibrin, Factors V, XIII, protein C, factor VII, platelets etc. thereby helping the clinical reader and physician - not being a specialist in molecular pharmacology - to understand the mechanisms and the potential clinical consequences.

In summary the potential or meanwhile witnessed clinical relevance of the pharmacological statin`s "effect on" or "interaction with" the clotting system (thereby clearly distinguishing between laboratory and clinical effects) should be presented in a clear and comprehensive way to attract readers and citations.  

I am sure that you will easily be able to meet these issues for improving tranparency of this manuscript and thereby attract readers.

Comments on the Quality of English Language

I am not authorized to professionally judge on the English language. A professional translator might be engaged for finalization of the text.

Author Response

Many thanks for your review.
I have modified the main text according to your suggestions, in particular:
1 I added a flowchart figure to describe the systematic procedures on how the cited publications were selected.
2. I have added two tables that simply summarize the literature cited and discussed in the text. To attract Readers we have kept the tables as simple as possible by including only essential and attractive information
3. I have added a figure with comprehensive illustrations presenting the potential pharmacological mechanisms of interaction of statins with clotting factors

Best Regards

Carmine

Round 2

Reviewer 1 Report

Comments and Suggestions for Authors

The presented manuscript has been corrected in response to the suggestions. The authors have followed the recommendations of the reviewer. After the revision, the provided data and addition of the results became more clear. I would like to thank the authors for resubmitting the manuscript and explaining the obscure points from the previous version.

Reviewer 4 Report

Comments and Suggestions for Authors

Dear authors,

thank you very much for reflecting my comments and include the proposed flow charts and tables. Thereby the manscript has significantly been improved, now being more attractive for the readers.

with kindest regards

Bernhard Rauch